# Effects of Twins Therapy on Egocentric and Allocentric Neglect in Stroke Patients: A Feasibility Study

**DOI:** 10.3390/brainsci13060952

**Published:** 2023-06-14

**Authors:** Woo-Hyuk Jang, Hyeong-Min Hwang, Jae-Yeop Kim

**Affiliations:** Department of Occupational Therapy, Kangwon National University, Samcheok 25949, Republic of Korea; wlqtksek@hanmail.net (W.-H.J.); gudalsv@naver.com (H.-M.H.)

**Keywords:** egocentric neglect, allocentric neglect, stroke, rehabilitation, twins therapy

## Abstract

(1) Background: Existing treatment methods for neglect are concentrated on egocentric neglect and may lead to various problems such as cost/space constraints and portability. Therefore, this study seeks to determine how a new treatment (also known as twins therapy, TT) for stroke patients can improve an existing problem associated with neglect. (2) Method: A pre/post-test control group research design was used and both groups continued to receive existing rehabilitation treatment, whilst TT intervention was only added to the experimental group. TT intervention was conducted for a total of 20 sessions (1 session for 30 min/day, 5 days/week, for 4 weeks). (3) Result: There was no significant difference in the manual function test (MFT) and the Korean version of the Modified Barthel Index (K-MBI) items (*p* > 0.05) before and after the TT intervention. However, the score and execution time of the apple cancellation test showed a significant reduction only in the experimental group (*p* < 0.05). (4) Conclusion: TT not only improved egocentric neglect, but also allocentric neglect symptoms in stroke patients.

## 1. Introduction

Neglect is described as a lack of spatial attention and is a symptom of impaired perception, attention and behavior in the opposite space of the damaged cerebral hemisphere, despite no sensory damage [1,2,3]. Neglect is caused by various pathological conditions, especially following a cerebral infarction or cerebral hemorrhage [2]. The incidence of neglect is more common in stroke patients of the right cerebral hemisphere than in stroke patients of the left cerebral hemisphere [3]. Symptoms of neglect caused by right cerebral hemisphere lesions mainly include neglect of the left space, impaired concentration and restriction of behavior on the transverse plane [4]. A limitation of response to the neglect aspect negatively affects body balance, functional ability such as walking and activities of daily living [5].

Neglect is classified into various subtypes according to the criterion (aspect, range and frame of reference, etc.) because of the heterogeneous symptoms [6]. Neglect in this study is classified into the following two subtypes depending on how the object is recognized based on the frame of reference: egocentric neglect (or viewer-centered neglect) and allocentric neglect (or object-centered neglect) [6]. Egocentric neglect is associated with dorsal visual pathways (involved in spatial perception) [7] and shows a defect in the ability to recognize objects or body parts, such as the head, trunk and arms, on the left side of the patient’s midline [1]. In contrast, allocentric neglect is associated with ventral visual pathways (involved in the visual identification of objects) [7] and has a defect in the ability to perceive the left side as the center of an object, regardless of the object’s location [1].

Various methods have been introduced to treat such various symptoms and types of neglect [8,9,10,11,12,13]. First, Pierce et al. (2002) classified the treatment methods for neglect into three main categories [8]: (1) treatments targeting arousal deficits; (2) treatments targeting deficient visual attention; and (3) treatments targeting spatial representation deficits. Treatments targeting the arousal deficit approach are a method of using drugs, such as anti-dopamine, to treat neglect [8]. Treatments targeting the deficient visual attention approach use a computer-based scanning program to increase attention regarding the patient’s space, while treatments targeting the spatial representation deficit approach use prism and trunk rotation to help reconstruct the neglected space that the patient is seeing [8]. On the other hand, Barrett et al. (2006) classified treatment methods into two main categories [9]. The first category includes the top-down and bottom-up approaches [9]. The top-down approach involves scanning training, which uses tactile and visual assist devices to urge patients to turn their bodies around and look at the left-hand space [9]. The bottom-up approach involves training designed to enhance or rearrange the characteristics of perceived information by stimulating the negated hemispace through frequent viewing [9]. The second category includes endogenous and exogenous approaches. The endogenous approach is used to increase the tendency to view the left space without special equipment, such as limb activation therapy, while the exogenous approach is used to induce dynamic remapping through the use of special equipment such as prism lenses [9]. Finally, a treatment method using virtual reality has recently been announced [10]. Virtual reality therapy has no physical spatial constraints and can improve activities of daily living and spatial attention concentration even in chronic patients [11]. However, the majority of treatment methods are concentrated on egocentric neglect caused by problems in spatial perception, and studies that are separated from allocentric neglect caused by problems in object identification are insufficient [7,12].

Neglect evaluation, on the other hand, is mainly divided into a pencil–paper test, behavioral assessment, clinical observation and virtual reality assessment [13]. However, existing evaluation methods mainly target egocentric neglect and they are insufficient for addressing other types of neglect [14]. For this reason, existing treatment methods are concentrated on the treatment of egocentric neglect, such as prism treatment, transcutaneous electrical nerve stimulation (TENS), optokinetic stimulation (OKS), etc. [15,16,17]. Moreover, the study of Turgut et al. mentioned the need to distinguish between egocentric neglect and allocentric neglect treatments, because existing treatments, such as prism treatment and cueing paradigm training, have only been verified to improve egocentric neglect [18]. Accordingly, as a result of previous studies on differential diagnosis, the evaluation method primarily used an apple cancellation test that can distinguish between egocentric neglect and allocentric neglect [19]. The treatment method mainly used a virtual reality navigation task (VRNT) [20] and repetitive transcranial magnetic stimulation (rTMS) [21]. Research shows that treatment using VR helps to re-identify space and improve concentration, and rTMS is effective not only for egocentric neglect, but also for allocentric neglect [21,22]. However, existing treatment methods lead to cost problems due to the use of expensive equipment [23]. They also have both spatial constraints and portability problems due to the size of the equipment [23]. Therefore, a method that can solve these problems and treat allocentric neglect (problem of object identification) at the same time, away from treatment methods that are biased towards egocentric neglect (problem of spatial perception), is highly necessary. For this, it is necessary to study an approach in which the arrangement of stimuli in the treatment background (EX, A4 paper) considers both the concept of spatial perception and the concept of object identification. 

To this end, the training method used in this study has two main features. First, in order to improve egocentric neglect, it is necessary to provide spatial exploration opportunities through a distributed placement of the stimuli to be selected. Therefore, more exploration and better focus on the left side, centered on one’s own body, would be possible. Next, to improve allocentric neglect, the distributed stimuli should be presented in pairs rather than singly, and only the same pair of stimuli should be selected. This should provide the opportunity to identify each object. Thus, it should be possible to improve object identification regardless of the location of the stimuli. In other words, we hypothesize that egocentric and allocentric neglect can be treated simultaneously via this training method. In addition, we expect that the change in attention via the improvement in these two neglects will affect the upper extremity function and activities of daily life.

Therefore, in this study, we sought to evaluate the feasibility of a new treatment (twins therapy, TT) that can simultaneously treat the concept of spatial perception and concept of object identification in addition to improving the existing problems (cost, space constraints and portability). Through this, we attempted to determine how TT affects the upper extremity function and activities of daily living, as well as simultaneous improvement in egocentric neglect and allocentric neglect in stroke patients.

## 2. Materials and Methods

### 2.1. Participants

This study included 15 people with both egocentric and allocentric neglect attributed to stroke, and used the Edinburgh Handless Inventory (EHI) to target 15 people with right dominant hands [24]. A sufficient explanation was provided to the participants, and the study started after written informed consent was obtained. The participants were randomly divided into 7 in the experimental group and 8 in the control group. The research design was based on a pre/post-test control group design and the selection criteria for all of the research participants were as follows: (1) those whose stroke lesion was diagnosed more than 3 months before; (2) those with symptoms of egocentric and allocentric neglect at the same time; (3) those with a score of 24 or higher in MMSE-K; and (4) those who understood and agreed with the purpose of the study. This study was approved by the Kangwon National University Institutional Review Board (KWNUIRB-2021-05-011-003) prior to commencement. 

### 2.2. Materials

#### 2.2.1. Korean Version of the Mini-Mental Status Examination (MMSE-K)

MMSE-K is an evaluation tool standardized by Kwon and Park (1989) to use the existing MMSE for the elderly in consideration of the educational background [25]. MMSE-K consists of seven items. Out of a total of 30 points, 24 points or more are classified as definite normal, 23~20 points are classified as suspected dementia and 19 points or less are classified as definite dementia. The inter-inspector reliability is 0.99 [25]. 

#### 2.2.2. Apple Cancellation Test

The apple cancellation test was developed by Bickerton et al. to differentiate between egocentric and allocentric neglect [19]. A total of 150 apples are randomly scattered on the test paper. One third of the apples are intact and two thirds are left or right side open. The test paper is divided into 10 zones (4 on the left, 2 in the middle and 4 on the right) due to two invisible rows and five columns. The participant proceeds by marking only intact apples, regardless of size. Each section contains 15 apples (3 large apples and 12 small apples), of which 5 are correct (1 large apple and 4 small apples). The test lasts up to 5 min and has a total score of 50 points. The analysis showed that egocentric neglect is divided into left-side egocentric neglect if it is a positive integer, and right-side egocentric neglect if it is a negative integer when subtracting the number of intact apples selected from the four areas on the left from the number of intact apples selected in the four areas on the right. Allocentric neglect counts the number of wrong answers (apples with left or right open) in the entire domain (including the middle 2 zones). When the number of apples with the left open is subtracted from the number of apples with the right open, it is divided into left-side allocentric neglect if it is a positive integer and right-side egocentric neglect if it is a negative integer [19,26]. The cut-off scores are ±2 for egocentric neglect and ±1 for allocentric neglect [23]. In this study, we used separate scores and execution times for egocentric and allocentric neglect, and a decrease in score indicated an improvement in symptoms.

#### 2.2.3. Manual Function Test (MFT)

MFT was developed by the Myeongja branch of the Faculty of Medicine, Tohoku University, Japan, and is a tool for assessing motor control and function of the upper extremity in stroke patients [27]. The MFT consists of 4 items of upper limb movement, 2 items of grip and 2 items of finger manipulation, and out of a total of 32 points, each sub-item is awarded 1 point when performed, and 0 points if it is impossible [27].

#### 2.2.4. Korean Version of the Modified Barthel Index (K-MBI)

K-MBI is an evaluation tool that was translated and standardized by Jung et al. (2007) in 2007 from the MBI (5th edition), published by Shah et al. in 1989 [28]. The questions consist of 10 assessment items, and can be divided into 7 self-care activities and 3 mobility activities [29]. The scores are divided into 5 levels with a total of 100 points being completely independent, 99~91 points being minimally dependent, 90~75 points being slightly dependent, 74~50 points being partially dependent, 49~25 points being maximum dependent and 24~0 points being completely dependent [30].

### 2.3. Procedure

The intervention used in this study is called twins therapy (TT), and consists of 7 sheets of A4 paper, each with 45 different pairs of stimuli (pictures, numbers and letters). The training procedure was to position the paper directly in the center of the patient, and then ask them to find 15 complete pairs of stimuli (targets, also known as twins) out of 45 pairs of stimuli. For spatial perception training for egocentric neglect, paired stimuli were arranged irregularly as if scattered on A4 paper. Then, instructions were given to find the complete pairs (twins) in this arrangement. For object identification training for allocentric neglect, patients were also instructed to select only complete pairs (twins), excluding stimuli from different pairs (distractor). 

Each sheet of paper means one step, it consists of a total of seven steps and various stimuli are presented as the level increases. Each step was presented in the order of numbers and letters, starting with shapes, and a mix was presented in steps 6 and 7. Starting from step 1, as the session was repeated, we encouraged them to progress to higher steps. In order to prevent an increase in the patient’s burden, we did not suggest a separate standard for going up to the next step and allowed them to proceed freely. Both groups continued with their existing rehabilitation, with only the experimental group receiving additional TT. TT was conducted for a total of 20 sessions (1 session for 30 min/day, 5 days/week, for 4 weeks) (Figure 1). All tests were performed twice, before and after TT training.

### 2.4. Data Analyses

In this study, the collected data did not show normality. The general characteristics of the subjects, such as sex, educational background and injury type, were analyzed using the chi-square test, and the age and duration of onset were analyzed using the Mann–Whitney U test to determine the mean value and significance level. We used the Mann–Whitney U test to assess homogeneity between the two groups and to compare the amount of change between the two groups, and Wilcoxon’s signed-ranked test for the pre/post comparisons.

## 3. Results

### 3.1. General Characteristics of the Participants

There were no significant differences between the two groups of participants regarding age, gender, level of education, type of cerebral damage, and onset duration (Table 1).

### 3.2. Baseline Similarity between Two Groups before Training

There was no significant difference in all tests, including the apple cancellation test (*p* > 0.05) (Table 2).

### 3.3. Comparison of Test Scores between the Experimental and Control Groups

In the experimental group, the apple score (Ego) and apple score (Allo) yielded significant results (*p* < 0.05). However, in the control group, there was no significant difference in all tests (*p* > 0.05) (Table 3).

### 3.4. Comparison of the Amount of Change between the Experimental and Control Groups

The comparison of the amount of change showed no significant difference in all scores, but there was a numerically greater improvement in the mean values of the experimental group for all scores (Table 4). In particular, the all apple score (Ego, Allo, sec) and MBI score showed more changes in the experimental group compared to the control group. Additionally, in the experimental group, the amount of change in the allocentric test score was higher than in the egocentric score. The detailed score change for each subject is presented in Appendix A.

## 4. Discussion

Treatment methods for neglect have been combined with cutting-edge technologies such as virtual reality (VR), transcranial magnetic simulation (TMS) and optic simulation (OKS) [15,16,17]. However, these studies have failed to consider the various types of neglect and spatial constraints have been identified due to the size and high cost of equipment [17,18,19,20,21,22,23,25,26]. Accordingly, this study was conducted to determine how the newly developed therapy, twins therapy (TT), affects stroke patients with neglect by improving these shortcomings.

First, the two groups are judged to be homogeneous groups with no differences in general characteristics and baseline similarity.

Next, pre/post comparisons within the group showed significant differences in both the egocentric and allocentric outcomes of the apple cancellation test, only in the experimental group. This means that twins therapy is effective not only in the treatment of egocentric neglect, but also in the treatment of allocentric neglect. There was no significant difference between the execution time of the apple cancellation test, manual function test (MFT) and the Korean version of the Modified Barthel Index (MBI). However, in all scores, smoother performance was confirmed in the experimental group. In particular, in the Apple_time (sec), there was a 12 s decrease in the control group, and a 60 s decrease in the experimental group. In addition, we believe that a training period of one month is relatively short to test our hypothesis that an improvement in neglect via TT will also affect the upper extremity function and activities of daily living.

Comparisons of the amount of change for each test item between groups showed no significant difference in all items. However, the amount of change in the test scores of all items was higher in the experimental group compared to the control group. In particular, the apple score (ego), apple score (allo) and apple time (sec) showed more changes in the experimental group compared to the control group. The amount of change in allocentric neglect was also higher than in egocentric neglect within the experimental group. In addition, in the Korean version of the Modified Barthel Index (MBI) category, the experimental group showed more change than in the control group. These preliminary results indicate the interest in further examining the effect of TT on larger samples of patients with the expectation that this therapy may have a positive effect on the functioning of activities of daily living as well as on neglect (especially in allocentric neglect). On the other hand, there was no significant difference in the control group, but the average score improved. Although TT was not performed, the improved score is attributed to the effect of the existing rehabilitation treatment.

The limitations of this study are as follows. First, it is difficult to generalize due to the small number of participants. Second, a single check of baseline similarity was not sufficient to differentiate between the patient’s own functional recovery before training and the effect after training. Third, there was no additional comparison of MBI subdomains or a more detailed comparison analysis of the proximal and distal extremity of MFT. Finally, it was not clear how the level of difficulty in each stage of TT was set. Future studies are likely required to resolve these limitations. Nevertheless, this study is significant because the newly developed twins therapy provides an opportunity to break free from the limitations of cost and space, which are limitations of conventional therapies. It is also worth noting that it is the world’s first pencil and paper training that can simultaneously improve the symptoms of egocentric neglect and allocentric neglect, an area that is still lacking in training methods.

## 5. Conclusions

This study sought to determine how twins therapy (TT) affects neglect in stroke patients. The results showed a significant difference in improving symptoms of egocentric neglect and allocentric neglect in the experimental group, but no significant difference was found in the MBI and MFT tests. Furthermore, there was no significant difference in all items when comparing the changes between groups, but the experimental group showed higher changes than the control group. Therefore, we confirmed the feasibility of TT. Through this study, we hope that TT will be used in more studies in patients with unilateral neglect in actual clinical practice.

## Figures and Tables

**Figure 1 brainsci-13-00952-f001:**
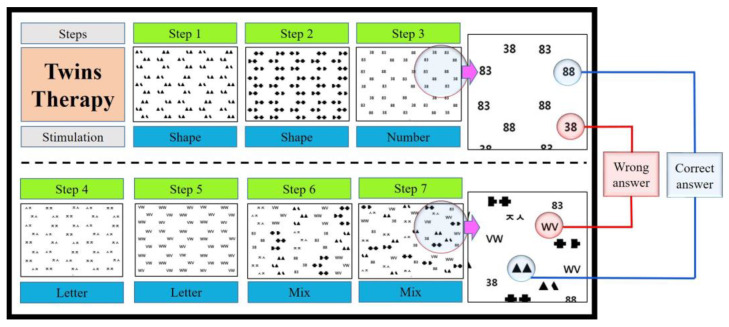
Steps and composition of Twins Therapy.

**Table 1 brainsci-13-00952-t001:** General characteristics of the experimental and control groups of patients (*N* = 15).

Characteristic	Classification	Experimental (*n* = 7) Mean ± SD	Control (*n* = 8) Mean ± SD	*p*
Age (year)	Average age	61.57 ± 16.33	62.37 ± 12.99	1.00
Gender	Male	3	5	0.447
	Female	4	3	
Education level	None	0	1	0.635
	Elementary	1	2	
	Middle school	1	1	
	High school	4	3	
	College	0	1	
	University or higher	1	0	
Damage type	Cerebral hemorrhage	5	5	0.714
	Cerebral infarction	2	3	
Duration of onset (day)		121.71 ± 41.70	119.12 ± 76.59	0.867

SD = standard deviation.

**Table 2 brainsci-13-00952-t002:** Baseline similarity between experimental and control groups.

Assessment Type	Experimental (*n* = 7) Mean ± SD	Control (*n* = 8) Mean ± SD	*p*
Apple score (Ego)	6.00 ± 3.36	9.00 ± 4.47	0.336
Apple score (Allo)	9.42 ± 4.42	7.12 ± 2.58	0.336
Apple_time (sec)	555.14 ± 137.85	542.00 ± 131.83	0.694
MFT_Rt	29.57 ± 1.39	28.25 ± 1.48	0.094
MFT_Lt	5.28 ± 6.89	4.62 ± 6.67	0.867
MBI	31.57 ± 12.35	38.75 ± 9.60	0.152

SD = standard deviation, Ego = egocentric neglect, Allo = allocentric neglect, sec = second, MFT = manual function test, Rt = right, Lt = left, MBI = Korean version of Modified Barthel Index.

**Table 3 brainsci-13-00952-t003:** Comparison of test scores between the experimental and control groups.

Assessment Type	Experimental (*n* = 7) Mean ± SD			Control (*n* = 8) Mean ± SD		
	Pre	Post	*p*	Pre	Post	*p*
Apple score (Ego)	6.00 ± 3.36	3.14 ± 1.21	0.039 *	9.00 ± 4.47	7.62 ± 5.50	0.139
Apple score (Allo)	9.42 ± 4.42	4.85 ± 2.67	0.017 *	7.12 ± 2.58	5.62 ± 1.50	0.231
Apple_time (sec)	555.14 ± 137.85	494.28 ± 181.91	0.063	542.00 ± 131.83	529.25 ± 118.22	0.575
MFT_Rt	29.57 ± 1.39	29.85 ± 1.46	0.317	28.25 ± 1.48	28.25 ± 1.48	1.00
MFT_Lt	5.28 ± 6.89	5.71 ± 7.13	0.180	4.62 ± 6.67	4.87 ± 6.89	0.157
MBI	31.57 ± 12.35	32.71 ± 11.71	0.066	38.75 ± 9.60	39.12 ± 9.47	0.180

SD = standard deviation, Ego = egocentric neglect, Allo = allocentric neglect, sec = second, MFT = manual function test, Rt = right, Lt = left, MBI = Korean version of Modified Barthel Index. * *p* < 0.05.

**Table 4 brainsci-13-00952-t004:** Comparison of test score variations between the experimental and control groups.

Assessment Type	Experimental (*n* = 7) Mean ± SD	Control (*n* = 8) Mean ± SD	*p*
Apple score (Ego)	−2.85 ± 2.26	−1.50 ± 3.11	0.536
Apple score (Allo)	−4.57 ± 2.29	−1.50 ± 3.20	0.094
Apple_time (sec)	−60.85 ± 69.64	−12.75 ± 53.50	0.189
MFT_Rt	0.28 ± 0.75	0.0 ± 0.53	0.694
MFT_Lt	0.42 ± 0.78	0.25 ± 0.46	0.867
MBI	1.14 ± 1.21	0.37 ± 0.74	0.232

SD = standard deviation, Ego = egocentric neglect, Allo = allocentric neglect, sec = second, MFT = manual function test, Rt = right, Lt = left, MBI = Korean version of Modified Barthel Index.

## Data Availability

The data used for the study are private but can be made available upon reasonable request.

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
