# Peer review of "Effects of Twins Therapy on Egocentric and Allocentric Neglect in Stroke Patients: A Feasibility Study"

_brainsci, 2023, doi:10.3390/brainsci13060952_

Round 1

Reviewer 1 Report

General comments

The paper describes a study on the recovery of egocentric and allocentric neglect after a treatment named Twins Therapy (TT).  The idea to investigate the efficacy of treatment over these two forms of neglect is interesting and potentially useful.  In my opinion, the study and the paper suffer from several important limitations.  First, the logic underlying the proposed treatment is not clearly spelled out by the authors.  Why should TT work?  The authors are not explicit about whether TT was originally meant for egocentric or allocentric neglect; I find it difficult to understand how TT can be meant to modify two different forms of neglect (which presumably are linked to the derangement of different mechanisms).  At any rate, if this is the hypothesis of the authors, it should be spelled out more clearly.  At present, there is no clear indication of the motivations to develop TT.  Second, there is an important issue with results.  In the Results section, it is clear from Table 4 that the experimental and control groups are not different in the two Apples variation scores.  Then, after ad hoc analyses presented in the Discussion section, the authors state that TT is effective and also has a positive effect on the function of activities of daily living (line 245; but, also for this test, the effect is not significant in Table 4).  I realize that there is some tendency in the data in the direction of the experimental group being sensitive to the treatment, but the discussion must consider all the formal analyses of findings not picking up only the more promising ones.  Of course, it would be particularly helpful if the authors could collect data from additional patients so as to make the results more solid.  Third, I think the procedure of the new treatment should be spelled out more clearly (see comments below).

In the form I have received of the manuscript, there was no reference section (even though the text reports expected numbering); thus, I cannot evaluate the appropriateness of the list of references.

Specific points

Line 26 Walkin should be walking

Line 28: “Neglect is classified into two types…”

There are actually various forms of neglect: extrapersonal, peripersonal, and personal is one distinction.  Far and near is another one, etc.

It would be clearer to make the point that neglect can be fractionated in various ways and that the present paper focuses on the egocentric-allocentric distinction.

These distinctions are also relevant as they were at the base of the various forms of treatment described in lines 35 and below.

Line 47: “The bottom-up approach is a method that…”

I don’t think this expression is clear.  The bottom-up approach is not a method in itself.

Similarly, in Line 50: “The endogenous approach is a method to increase…”

Line 50. I think the authors by “classification” mean “category”.

Line 60. “However, existing evaluation methods are mainly targeting egocentric neglect and it is insufficient to address other types of neglect [14].”

I guess it would be more appropriate to say “… and they are insufficient”.

Note that here authors refer to “other types of neglect”, i.e., more than two, even though they do not state the forms of neglect they refer to.

Line 65. Turgut et al (2018)

The ref number should be inserted not the year.

Line 77. “Therefore, in this study, through a new treatment (Twins Therapy, TT)…”

There is no indication in the Introduction about what the TT is, how is motivated, etc.

It is only said that is “new”.  This does not represent sufficient motivation for the study.

The theoretical basis of this new therapy and its general characteristics should be spelled out in the motivation of the study.

Line 85. “The participants heard enough explanation about the study and conducted the study after receiving a written consent.”

This sentence is awkward.

Line 102.  Bickerton et al (2011)

Ref numbering should be used.

Figure 1.  The Apples test is a standard test. So, there seems to be a need to represent it in a figure.

Line 137.  The procedure of the TT should be more detailed.

What is the meaning of the steps?  Difficulty? Other?

When does a patient is moved from one step to the next one?  Are they threshold levels of performance?

As spelled above, something more should be said about the logic of this treatment.  What is its rationale? How is it connected with the egocentric and allocentric forms of neglect?

As stated above, I think this argument should be spelled out at the end of the Introduction when the treatment is originally proposed.  However, also here the presentation does not make sufficiently clear the theoretical bases of the proposed treatment.

Table 1.  There is some problem with the pagination of Table 1.  Check lines 164-167.

Line 187: “In the experimental group, the Apple score (Ego) and Apple score (Allo) were significant (p<.05).”

This sentence is unclear.  I think the authors refer to the pre-post differences in performance, but this should be said more clearly.

Line 186 and following

In part 3.3 the authors analyze the effect of treatment.

Their treatment of data is cautious as they use non-parametric analyses. However, they do not explicitly motivate this choice.  Did they check the distributions of raw data and found significant deviations from normality? 

I believe that Bickerton et al (2011) used parametric analyses.  So, the choice to use non-parametric tests is not necessarily obvious (although it may be the correct one).  At any rate, it should be at least motivated more explicitly.

Lines 237 and following: “In particular, the Apple score (ego), Apple score (allo) and Apple time (sec) showed more changes…”

Analyses should be in the Results section not in the Discussion section.

Furthermore, these informal analyses in the Discussion should be commented on jointly with the analyses presented in the Results section.  Table 4 reports that there was no difference between the two groups either in the Apple score (ego) and Apple score (allo).  This is something that cannot be simply ignored.

Line 254: “It is also worth noting that it is the world's first pencil & paper training that can simultaneously improve the symptoms of egocentric neglect and the improvement of allocentric neglect, which is still lacking in training methods.”

As egocentric and allocentric neglect are distinct and presumably based on different mechanisms I do not share the positive opinion of the authors on the idea that one single training can be effective with both.  In part, as stated above, there is no clear-cut evidence that TT is effective.  At any rate, the authors should be more explicit on what is the rationale of TT and what it really aims to. 

The text is generally clear.  However, it would be certainly useful to have the manuscript checked by an English native speaker.

Reviewer 2 Report

This is an interesting pilot study testing the efficacy of a new intervention tool for neglect that may target both allocentric and egocentric neglect which is low tech and, therefore, inexpensive and doesn't have a lot of physical space requirements.  The intervention group scores were compared to the control group scores after 20 sessions.  There are several recommendations for enhancing this report:

1) It is not clear how this intervention is different than the commonly used, and long-existing, task practice of visual scanning and cancellation type tasks used clinically and in some research, other than the participants find an identical figure.  The authors should describe why they think this intervention tool is better than these existing ones.

2) The figure showing the intervention would be easier to understand if the reader could see the right answer to show what the participant had to do.

3) line 83 - did the participants  have to have both egocentric and allocentric neglect or they could have one or the other?

4) line 53 - I'm not sure what you mean by "no space constraints" - are you talking about physical space or the idea that VR can be very expansive within the virtual space about space that is shown?

5) line 85 - I'm not sure what you mean by "heard enough explanation" here.  To do what?

6) line 87 - were participants randomly assigned to groups or just non-randomly assigned using this software - as this is a statistical software package, I'm not sure how you assign participants with it. Please explain.

7) Line 125 - the term "mobility" usually refers to moving the whole body through space, as in walking or with a wheelchair, so it sounds odd when talking about the arm.  Do you mean motor control and function of the UE?

8) section 3.4 and the paragraph starting line 235:  I'm not sure what you mean by "test score variations".  Do you mean change scores?   If so, usually people refer to this as "amount of change" so I'm suggesting that you use that term.  If not that, please explain.

9) Results: 

            a. Is there an MCID for the Apples scores?  Some of the change                        scores are really small so I'm not sure they are really meaningful.  

            b. "homogeneity test" - I think you mean testing for baseline                              similarity between groups, right? Usually this is the language that                  is used for this construct and much clearer to understand so I'd                      suggest using it for readers.

10) Discussion - The complicated issue is that the within group analysis showed significant, with unclear meaningfulness right now, change for the experimental group but not for the control group. Yet the change scores (I'm assuming the variation table are change scores) were not different between the groups.  This argues that there must have been enough variability within the groups in their improvement that they can't be called separate groups numerically. This then makes it difficult to say that really only one of the groups improved.  Now, maybe this is a power issue as the sample size was small.  However, it seems a little ingenious to state that only one group improved. I think one thing that would make it clearer is if you could talk about the number of participants that improved by some meaningful amount in each group  or even give the change scores for each participant (15 participants is not that huge to make that impossible). That would give clearer ideas about the range of changes within each group and if some of this overlap is only due to 1 participant as an outlier or so.  

The major issues with the English are detailed above and are mainly some word choices which are not the common way the field refers to some things.  

Reviewer 3 Report

It was a pleasure to read the paper.

I would like you to expand cooperation with more institutions in your country and abroad.

I recommended your work to the editor for the press.

Round 2

Reviewer 1 Report

General comment

I think the revision has improved the paper and has dealt with some of the critical issues raised in my previous comments.  Still, I think that there are several areas that need considerable work before the paper can be appropriate for publication.  Below I detail various comments on specific points. Here, I would like to underscore two general points which I think are particularly critical.  First, the authors should be more explicit on the hypotheses underlying TT therapy.  What is the logic with which it was devised?  Why should it be effective with both egocentric and allocentric neglect?  I find that in this respect the paper is still deficient.  The authors have developed a new form of intervention against neglect.  They should explain why they think it might work and how.  Second, in the revision, the authors added the expression “A feasibility study” to the title.  Thus, they underscore the initial value of this study and the need for further research on TT therapy before a definite conclusion on its effectiveness can be reached.  This is fine, particularly in view of the observation that some of the critical results do not reach significance.  I would make sure that this point is clearly spelled out throughout the manuscript.  In particular, presently this is the case in the discussion section.

Specific comments

Line 14.  “how treatment”.

Maybe “how a new treatment”…

Line 20: MFT and MBI are not defined in the abstract

Line 79: “…and they are insufficient”

The sentence does not close. I would maintain the now canceled part “to address other types of neglect”

Line 100 “we sought to confirm the feasibility of a new treatment…”

Perhaps: “we sought to evaluate the feasibility…

Line 100: “that can simultaneously treat the concept of spatial perception and concept of object identification in addition to improving the existing problems.”

I find this sentence unclear in two respects.

First, the idea that TT can treat both egocentric and allocentric neglect is a hypothesis to be tested and this should be said more clearly. Also, it should be spelled out which is the basis for this hypothesis.  Why do the authors expect TT to have this simultaneous influence? Which are the general characteristics of TT that suggest such a hypothesis?  This is an important point to develop more explicitly in a revision.

Second, I find unclear the end of the sentence: “in addition to improving the existing problems”.  To which “existing problems” do the authors refer?

Line 102.  “Through this, we attempted to determine how TT affects the upper extremity function and activities of daily living as well as simultaneous improvement of egocentric neglect and allocentric neglect in stroke patients.”

This sentence states a fact as the authors indeed performed these tests.

However, what seems to be missing is the motivation why a training supposedly aimed to improve egocentric and allocentric neglect should also ameliorate upper extremity function.  I am not saying that this is not plausible.  However, this should be stated explicitly. 

Note also that one could see these tests from two different perspectives.  First, one could anticipate that TT therapy has a selective influence on neglect; accordingly, improvement in extremity function is not expected and the MFT can be seen as a control test.  As an alternative, one can hypothesize that by altering visual exploration TT might also improve daily life activities and possibly also motor control (which has been shown to be sensitive to attentional control).

So, different hypotheses can be put forward.  Which do the authors prefer and refer to?

Of course, this has an influence on the comments on the results.  A non-significant difference is expected for a control test but not for a test for which a change is expected.

All in all, the authors should be more explicit about their hypotheses.

2.2.2. Apple cancellation test

Reference to Figure 1 is not indicated in the text.  If the authors want to present this figure (which to me is not strictly necessary) they should ask permission from the original authors (Bickerton et al.) and Journal (Neuropsychology), as this is not an open-source journal.

Line 200. “We used the Mann-Whitney U test for homogeneity between the two groups and the Mann-Whitney U test to compare the amount of change between the two groups,…”

Mann-Whitney U test is repeated and the sentence can be simplified:

“We used the Mann-Whitney U test for homogeneity between the two groups and to compare the amount of change between the two groups,…”

Line 207.  “There were no significant differences among the participants in each group and the general characteristics were as follows  (p>.05) (Table 1).”

The sentence is unclear.  Maybe:

“There were no significant differences between the two groups of participants for age, gender, level of education, type of cerebral damage, and onset duration (Table 1).”

Line 211. “General Characteristics (N=15)”

The table presents the characteristics of the two sub-groups not of the general group of patients.

Maybe: “General Characteristics of the experimental and control groups of patients”.

Line 219.  The N seems unnecessary; delete the parenthesis.

Same on line 245 (legend of Table 4)

Line 237.  “… greater improvement in the mean value of the experimental group in all items (p>.05) (Table 4). “

Maybe: “… numerically greater improvement in the mean values of the experimental group for all items (Table 4).“

Probably, the term “items” should be “scores”.

Line 242.  “is presented in the appendix (Appendix).”

Change to “is presented in the Appendix”.

Line 248.  Discussion section

The arguments of the discussion are diluted.  The first paragraph repeats the aims of the study and the next two repeat  the results.  In general, these paragraphs are not very informative (particularly the second one starting on 257) and should be shortened.  The discussion should really focus on the meaning of the findings not simply on repeating the results.

Line 257.   Delete “15”.

Line 267. “However, the execution time of the Apple Cancellation Test, which cut about 12 seconds in the control group, and about 60 seconds in the experimental group, confirmed smoother performance.”

The sentence does not read well.  Please check for the English style.

Line 272. “no significant difference in all items.”

Items or scores?

Line 274 and following.

The presentation of actual numbers should be avoided in the Discussion section.  All the presented numbers can be retrieved in the Tables.  The focus should be on commenting on the results not on repeating them.

It is important to stress that the presentation concerns differences which are not significant.  Therefore, it is critical that this point is made very clear and that it is made plain that these are only exploratory comments.

The authors seem aware of this aspect:

“This suggests that TT also has a positive effect value on the function of activities of daily living as well as on neglect (especially in allocentric neglect) and is expected to result in significant differences in future studies with more participants.”

Still, it should be more explicit that these are hypotheses based on the present preliminary non-significant results and are only suggestive of future research (cannot be used for conclusions, at present).  For example:

“These preliminary results indicate the interest in further examining the effect of TT on larger samples of patients with the expectation that this therapy may have a positive effect on the functioning of activities of daily living as well as on neglect (especially in allocentric neglect).”

I am not myself an English native speaker.  So, my comments are not critical on this point. At any rate, I think the paper would greatly benefit from editing by an English native speaker.

Reviewer 2 Report

I am satisfied with all the revisions made to this paper